# The Microbiome of Peri-Implantitis: A Systematic Review and Meta-Analysis

**DOI:** 10.3390/microorganisms8050661

**Published:** 2020-05-01

**Authors:** Philipp Sahrmann, Fabienne Gilli, Daniel B. Wiedemeier, Thomas Attin, Patrick R. Schmidlin, Lamprini Karygianni

**Affiliations:** 1Clinic of Conservative and Preventive Dentistry, Center of Dental Medicine, University of Zurich, Plattenstrasse 11, CH-8032 Zurich, Switzerland; fabienne.gilli@bluewin.ch (F.G.); thomas.attin@zzm.uzh.ch (T.A.); patrick.schmidlin@zzm.uzh.ch (P.R.S.); lamprini.karygianni@zzm.uzh.ch (L.K.); 2Statistical Services, Center of Dental Medicine, University of Zurich, Plattenstrasse 11, CH-8032 Zurich, Switzerland; daniel.wiedemeier@zzm.uzh.ch

**Keywords:** culture-dependent techniques, hybridization, oral pathogens, PCR, pyrosequencing

## Abstract

This review aimed to systematically compare microbial profiles of peri-implantitis to those of periodontitis and healthy implants. Therefore, an electronic search in five databases was conducted. For inclusion, studies assessing the microbiome of peri-implantitis in otherwise healthy patients were considered. Literature was assessed for consistent evidence of exclusive or predominant peri-implantitis microbiota. Of 158 potentially eligible articles, data of 64 studies on 3730 samples from peri-implant sites were included in this study. Different assessment methods were described in the studies, namely bacterial culture, PCR-based assessment, hybridization techniques, pyrosequencing, and transcriptomic analyses. After analysis of 13 selected culture-dependent studies, no microbial species were found to be specific for peri-implantitis. After assessment of 28 studies using PCR-based methods and a meta-analysis on 19 studies, a higher prevalence of *Aggregatibacter actinomycetemcomitans* and *Prevotella intermedia* (log-odds ratio 4.04 and 2.28, respectively) was detected in peri-implantitis biofilms compared with healthy implants. *Actinomyces* spp., *Porphyromonas* spp. and *Rothia* spp. were found in all five pyrosequencing studies in healthy-, periodontitis-, and peri-implantitis samples. In conclusion, the body of evidence does not show a consistent specific profile. Future studies should focus on the assessment of sites with different diagnosis for the same patient, and investigate the complex host-biofilm interaction.

## 1. Introduction

To date, dental implants have shown high survival rates of up to 99% over 10 years [1,2]. Even if much stricter criteria on success are applied, the concept of dental implantology still appears promising [3,4], despite the fact that certain limitations of the relevant techniques become evident. Besides minor prosthetic complications (such as crown loosening or ceramic chipping, which can mostly be resolved easily, and without big effort) peri-implantitis, as the most common reason for biologic failure, is much more challenging [4,5]. The prognosis of peri-implantitis therapy, however, is far away from satisfactory today [6,7]. The key feature of peri-implantitis is the progressive loss of marginal peri-implant bone as a symptom of chronic inflammation of the peri-implant tissues [6]. While particular co-factors, such as diabetes mellitus [8,9], tobacco smoking [10,11], and insufficient oral hygiene [12,13] were found to accelerate the progress of bone destruction, the primary etiologic reason for the inflammation of peri-implant tissues is the oral biofilm [14]. Peri-implantitis-associated biofilms, colonizing exposed implant surfaces, are composed of a plethora of microbial species [15,16]. Accordingly, any cause-related treatment of peri-implantitis aims at effectively removing established microbial biofilms and preventing new biofilm formation [6]. Like in the therapy of periodontal inflammation, which is also biofilm-induced, mechanical biofilm removal has been proven to be the most efficient treatment modality. Therefore, hand instruments, such as scalers or curettes, ultrasonic tips, or air-abrasive devices [6,17], and sometimes different kind of lasers [18], are used. Moreover, antimicrobials have also been used, yet with limited success [19,20]. Both measures, however, mechanical and antiseptic biofilm control, represent very generic treatment strategies and do not conform to the need for individualized, specific, and sophisticated therapy schemes. In general, both therapeutic approaches show strong limitations. In particular, mechanical access for debridement of submucosal areas and narrow infra-bony defects is difficult and cannot be performed to a satisfactory extent [17,21,22]. Antimicrobial agents, on the other hand, are largely ineffective due to the immanent defense mechanisms of mature biofilms, such as osmotic barrier function and downregulation of bacterial metabolism [23], especially if the biofilm infrastructure had not been destroyed previously by mechanical means [24,25]. Gaining more specific insights into the composition of peri-implant biofilm might trigger the development of targeted treatment approaches and, thereby, improve the prognosis of peri-implantitis treatment. Regarding the composition of peri-implantitis-associated biofilm, a number of key pathogens have been strongly associated with the peri-implant inflammation so far. Since suppuration is a characteristic clinical finding in cases of peri-implantitis, *Staphylococcus aureus* has been suspected to play a major role in the pathology of the disease due to its typical pyogenic potential, which is already well-known in the field of dermatology [26,27]. However, recent studies using modern diagnostic tools and molecular-based identification techniques did not support this hypothesis [28,29]. After comparing the microbiome of healthy and inflamed implant sites, Belibasakis and co-workers found a predominance of three groups of *Treponema* spp. and a *Synergistetes* cluster A around diseased implants [30].

However, there is still some controversy among researchers about whether the composition of biofilm in peri-implantitis is really different from the composition of biofilms in periodontitis-affected sites, or even from the microflora around healthy dental implants [29,31,32]. Accordingly, the aim of the present study is to systemically review the relevant literature regarding the composition of biofilm of diseased peri-implantitis sites and to compare the microbiome of healthy sites to that found at sites affected by periodontitis. Additionally, this systematic review aimed to describe the microbiologic profiles of peri-implantitis, periodontitis, and healthy implants based on culture-dependent and culture-independent methods.

## 2. Materials and Methods

### 2.1. Review Questions

This systematic review aimed to answer the following focused questions:Is there a difference between the microbiome associated with implants affected by peri-implantitis and that associated with periodontitis-affected teeth?Is there a difference between the microbiome associated with implants affected by peri-implantitis and that associated with clinically healthy implants?What are the microbial profiles of peri-implantitis, periodontitis, and healthy implants, based on culture-dependent and culture-independent methods, from peri-implantitis versus periodontitis, or peri-implantitis versus healthy sites?

Based on the available data, the main hypothesis of this review was that specific microbial species are present, either exclusively or predominantly at peri-implantitis sites, but not around healthy implants or teeth with periodontitis.

### 2.2. Study Eligibility

For literature search, studies assessing the microbiome of peri-implantitis in systemically healthy patients were considered if microbiological testing included more than one microbial species, and the assessment was based on at least 10 samples per study. For inclusion, clear diagnostic criteria for peri-implantitis were applied. Only original studies were considered while systematic and narrative reviews were excluded. Likewise, studies on in-vitro-biofilms, animal studies and studies on cohorts with specific predisposing factors for peri-implant or periodontal diseases were excluded.

### 2.3. Search Strategy

An electronic search was conducted in five databases (U.S. National Library of Medicine database (MEDLINE) available online: www.ncbi.nlm.nih.gov/pubmed, Experta Medica Database (EMBASE) available online: www.embase.com, SCOPUS available online: (www.scopus.com), Biosis (available online: www.library.ethz.ch/Ressourcen/Datenbanken/BIOSIS-Previews) and Web of Science (available online: apps.webofknowledge.com)). The search items were divided into the groups peri-implantitis, microbiome, and assessment method, and the search for the respective sub-items was combined with the Boolean operators (OR, AND). The search strategy is reported in detail in Appendix B Literature Search Strategy. The electronic databases were searched from the times of their establishment until May 2019, and the relevant studies were considered for inclusion. There was no language restriction for the literature search.

### 2.4. Study Selection

In the first step, titles and abstracts yielded from the electronic search were screened and checked for possible inclusion. In the second step, full texts were reviewed in order to decide for or against final inclusion. Both steps were performed by two authors (FG and PS) independently. In case of any discrepancies during title and abstract screening, the respective paper was included to the full-text assessment, while discrepancies during this latter step were resolved by discussion between the authors (PS and FG) conducting the literature search.

### 2.5. Data Organization

In order to collect and merge the information of the different studies retrieved, a standard document was utilized. In particular, data regarding authors and the year of publication, study type and design, diagnostic criteria, number and type of samples, statistical units, implant systems used, sampling methods, and the microbiological assessment methods were extracted. Likewise, quantitative data regarding the microbiome for cases of peri-implantitis, periodontitis, and healthy implants were extracted.

### 2.6. Data Quality Assessment

The extracted data were assessed according to the Preferred Reporting Items for Systematic Reviews and Meta-Analyses guidelines (PRISMA) [33]. For each of the included studies, a quality assessment was performed by one investigator (PS) using an adaptation of the modified Newcastle–Ottawa scale [34]. Studies were evaluated regarding the case definition, the selection of the controls and their matching for confounding factors, as well as the blinding of the involved laboratory. A maximum of 7 points could be reached, resulting in a rating of high quality (6–7 points), medium quality (3–5 points), and low quality (0–2 points).

### 2.7. Data Synthesis

In order to assess literature for consistent evidence of exclusive or predominant peri-implantitis microbiota, statistical comparisons were performed only for microbial species that were assessed and reported in the majority of studies. Results beyond comparable data were analyzed qualitatively in separate sections. Meta-analyses were performed if more than three studies assessing the same bacterial species with sufficient homogeneity were found. Otherwise, weighted means for the (positive) detection of different species for either peri-implantitis or healthy sites were calculated.

### 2.8. Statistical Analysis

Random-effects meta-analyses using odds-ratios (OR) were performed to investigate if certain species were more prevalent at healthy sites than at peri-implantitis sites and vice-versa. Diagnostic plots (funnel plot, radial plot, standardized residuals, and normal QQ plot) were used to check model assumptions and sources of bias (e.g., publication bias). All meta-analyses and corresponding figures were calculated with the statistical software R [R Core Team, 2018, a language and environment for statistical computing; Available online: https://www.R-project.org/], including the metafor package [35].

## 3. Results

### 3.1. Selected and Excluded Reports

After removal of duplicate titles from different electronic libraries, 956 titles were retrieved for possible inclusion. Of these, 158 full texts were analyzed and further 95 studies were excluded for different reasons, such as no assessment of peri-implantitis (*n* = 37), missing quantification of the results (*n* = 35), inclusion of less than 10 samples (*n* = 14), assessment of only one single bacterial species (*n* = 6), or other reasons (Figure 1). For the screening process inter-rater reliability was 96% before discussion. Finally, data out of 64 studies on 3730 samples from peri-implant sites were extracted over a publication period from 1990 to 2019. More than 70% of these studies had been published in the past eight years. In the selected studies, different assessment methods were described, namely bacterial culture, PCR-based assessment, hybridization techniques, pyrosequencing, and transcriptomic analyses. While culture-dependent (44%) and hybridization techniques (44%) were the most frequently applied microbiological techniques for the studies published prior to 2011, culture-independent techniques, e.g., PCR, hybridization, pyrosequencing, and transcriptomics (42%), were predominantly described in studies published after 2011. Quality assessment revealed 9 studies of high, 25 of medium, and 30 of low quality, respectively (Table 1). In 59 out of 64 studies, sampling was performed using sterile paper points and/or curettes in the vast majority of cases. Few data were available regarding the type of investigated implants. A total of 17 studies, however, reported microbiological data on titanium implants.

### 3.2. Culture-Dependent Techniques

In a total of 13 studies using culture techniques, *Porphyromonas gingivalis*, *Tannerella forsythia*, *Prevotella intermedia*, *Parvimonas micra*, *Fusobacterium nucleatum*, *Campylobacter rectus*, *Aggregatibacter actinomycetemcomitans*, *Eikenella corrodens* and *Candida albicans* were compared, and weighted means for detection frequencies were calculated. Nine studies analyzed solely the microbiome of peri-implantitis cases, while four studies compared the respective findings of peri-implantitis-associated versus healthy implants. The yeast *C. albicans* was the only species that could not be retrieved from the tested peri-implantitis samples, but solely from healthy implant sites. After the analysis of all selected culture-dependent studies, no microbial species were found to be specific or characteristic for peri-implantitis.

A considerable heterogeneity regarding the statistical unit, i.e., whether one or more samples per patient had been taken, and considerable differences in culture techniques was found. Accordingly, a reasonable meta-analysis was not possible in order to assess potential statistical differences regarding the positive detection of specific microbial species between health periodontal and peri-implant disease.

However, the use of weighted means from single studies to illustrate the overall results revealed that the majority of microbial species was detected more frequently in healthy sites, while *Eikenella corrodens* prevailed in peri-implantitis samples (Figure 2). The latter was reported in a high-quality study by Leonhard et al. (1999), in which samples from healthy and peri-implantitis sites of the same patient were directly compared. Table 2 provides an overview of reports assessing different microbial species using culture-dependent methods.

### 3.3. Hybridization Techniques

Fifteen studies describing hybridization techniques to assess different genera were retrieved. No meta-analysis could be performed for all relevant studies due to heterogeneity in the data presentation. Instead, the results of only twelve studies could be compared in terms of weighted means for positive detection frequencies. In all peri-implantitis samples, *Campylobacter* spp., *Fusobacterium* spp., *Gemella* spp., *Porphyromonas* spp., *Parvimonas* spp., *Treponema* spp. and *Veillonella* spp. could be identified, while *Prevotella* spp., *Staphylococcus* spp. and *Streptococcus* spp. were detected in 60%–80% of samples (Figure 3 and Table 3). No conclusive differences between samples from healthy implants or peri-implantitis could be found.

### 3.4. PCR-Based Assessment

Twenty-eight studies that used PCR-based methods were selected for assessment. Due to a considerable homogeneity of the studies, a meta-analysis could be performed to assess the prevalence of *A. actinomycetemcomitans*, *P. intermedia,* and *T. forsythia* based on five, six, and eight comparable studies, respectively. The species *A. actinomycetemcomitans* was found to be prevalent in peri-implantitis with a significantly higher log-odds ratio (4.04) as compared to healthy implants (Figure 4). Regarding *P. intermedia* (Figure 5), the log-odds ratio for peri-implantitis was also significantly higher (2.28) than for healthy implants, although meta-analysis diagnostics could also indicate some publication bias (funnel and Galbraith plot, Appendix A). For *T. forsythia,* the prevalence turned out to be only slightly higher for peri-implantitis, failing to show significance with a log-odds ratio of 1.70 (Figure 6), which was even lower (1.03) if one considered the study by Sato et al., 2011 [83] as an outlier. Data for *P. gingivalis*, *T. forsythia*, *Treponema denticola*, *P. intermedia*, *P. micra*, *Prevotella nigrescens*, *F. nucleatum*, *C. rectus*, *Eubacterium nodatum*, *A. actinomycetemcomitans*, *E. corrodens,* and *C. albicans* were assessed in the retrieved studies. Of these, 17 studies allowed for the comparison of weighted mean values for positive detection frequencies (Figure 7). Eleven studies compared results between healthy implants and implants affected by peri-implantitis, while two studies compared peri-implantitis with periodontitis samples. Four further studies analyzed samples only from peri-implantitis sites. Notably, none of the assessed microbial species was exclusively found in either peri-implantitis, healthy implants, or periodontitis samples. In exception for *P. nigrescens* and *E. nodatum*, which slightly prevailed in peri-implantitis samples, the prevalence of *P. gingivalis*, *T. forsythia*, *T. denticola*, *P. intermedia*, *P. micra*, *F. nucleatum*, *C. rectus*, *A. actinomycetemcomitans*, *E. corrodens,* and *C. albicans* was similar in health and disease (Table 4).

Although the study by Yu et al. [96] posed the same focused question as the present review, the presentation of the results data did not allow for the inclusion of the study in the meta-analysis. In this study, the vast majority of microorganisms found in both peri-implantitis and healthy implant sites were spirochetes, with a proportion of 97.6%. Likewise, a total of 31 so-called “core-species”, including the above-mentioned microbial species, were found in more than 90% of both peri-implantitis and healthy samples. Beta-diversity analyses revealed the largest variance on the subject level. Differences regarding health status of the respective sites were reflected by differences of operational taxonomic units (OTUs) of *Prevotella* spp., *Porphyromonas* spp., *Tannerella* spp. and *Treponema* spp. In an earlier report, Belibasakis et al., 2016 reported that *Treponema* cluster 1 and a cluster A of *Synergistetes*, a gram-negative anaerobic rod, were highly associated with peri-implantitis [30].

### 3.5. Pyrosequencing

Overall, the selected reports differed considerably from each other regarding their focused question, genera assessment, data presentation, and outcomes. Therefore, meta-analysis with data of the respective studies was reasonable only for a limited selection of sufficient homogeneity. In five studies, bacterial assessment was performed by pyrosequencing, allowing for positive detection of the following genera: *Actinomyces* spp., *Campylobacter* spp., *Fusobacterium* spp., *Gemella* spp., *Parvimonas* spp., *Porphyromonas* spp., *Prevotella* spp., *Rothia* spp., *Staphylococcus* spp., *Streptococcus* spp., *Treponema* spp., and *Veillonella* spp. Of these, *Actinomyces* spp., *Porphyromonas* spp., and *Rothia* spp. were found in all studies in samples from healthy sites, periodontitis sites, and peri-implantitis sites. The genera *Campylobacter* spp. and *Gemella* spp., likewise, were present in all samples collected from healthy sites. However, one report that did not assess samples from healthy controls failed to detect species of these genera in periodontitis or peri-implantitis samples [69]. Interestingly, *Fusobacterium* spp. and *Veillonella* spp. were detected in all studies investigating peri-implantitis and periodontitis samples except from one study [66], in which the respective species were not found in samples from healthy sites. Genera showing any specificity, i.e., complete absence or presence in either peri-implantitis or healthy implant or periodontitis samples, were not found (Table 5).

In a recent report [90], *Prevotella* spp., *Staphylococcus* spp., *Tannerella* spp., and *Treponema* spp., were detected among other genera in both healthy and peri-implantitis samples, whereas da Silva et al. [55] identified the aforementioned genera in neither healthy nor peri-implantitis samples. Nevertheless, the latter study reported statistically significant differences between certain genera, i.e., lower levels of *Actinobacteria* spp. and higher levels of *Bacterioidetes* spp. at peri-implantitis sites as compared to healthy implant sites.

In an earlier study, Tsigarida et al. [90] investigated, primarily, the effect of smoking on 79 different microbial species from 80 partially edentulous subjects with peri-implant health or disease, and identified high proportions of several putative pathogens in the biofilm of smokers. In general, peri-implantitis samples were reported to show a non-specified broader abundance of disease-associated microbial species than samples from healthy sites. Moreover, inter-individual discrepancies between the microbiome of peri-implantitis, and periodontitis within individual patients, were highlighted by Dabdoub et al. [56] and Kumar et al. [66], who confirmed the presence of a significantly broader microbial diversity in periodontitis compared to peri-implantitis. The group of Kumar et al. pointed out that previously unknown species were only identified in samples from peri-implantitis sites, while higher levels of *Actinomyces* spp., *Peptococcus* spp., *Campylobacter* spp., *Streptococcus* spp. and *Butyrivibrio* spp. were detected in biofilms from peri-implantitis as compared to those at healthy implants. However, Murayama et al. revealed the presence of higher proportions of *P. nigrescens*, yet lower proportions of *Peptostreptococcaceae* spp. and *Desulfomicrobium orale* in peri-implantitis than in periodontitis, respectively [69].

A recent study of Kröger and coworkers showed that the pocket depth has a relevant effect on the microbiome of the peri-implantitis sites, highlighting a pronounced dysbiosis in deeper pockets, while other species, *Acinetobacter* spp. and *Rhodobacteraceae* spp., were detected exclusively in pockets of less than 8 mm of depth [98].

### 3.6. Metatranscriptomic Analysis

In one single study [87], meta-transcriptomic analysis was applied to analyze the microbiome of peri-implantitis biofilms. The genera identified in twelve samples from healthy and peri-implantitis affected implant sites were *Actinomyces* spp., *Campylobacter* spp., *Fusobacterium* spp., *Gemella* spp., *Parvimonas* spp., *Porphyromonas* spp., *Prevotella* spp., *Rothia* spp., *Staphylococcus* spp., *Streptococcus* spp., *Tannerella* spp., *Treponema* spp., and *Veillonella* spp. (Table 6). However, the genera *Gemella* spp. and *Veillonella* spp. was not found in peri-implantitis sites.

### 3.7. Pathogen-Specific Analysis

A total of five studies aimed to specifically assess *S. aureus* in samples from peri-implantitis. Of these, one report compared peri-implantitis sites with healthy implant sites and another study described the difference between peri-implantitis and periodontitis sites (Table 7). One report highlighted the detection of *Archaea* spp., which were found four times more frequently in peri-implantitis samples than in samples from healthy sites (Table 8). Viruses were assessed in four studies. Human cytomegalovirus (CMV) was detected in much higher proportions in 2 out of 3 studies, while Epstein–Barr virus (EBV) was identified in peri-implantitis samples in all three studies. Herpes simplex virus (HSV) was assessed in one study, only in 33% of peri-implantitis sites, and 24% of healthy sites, respectively.

## 4. Discussion

In this systematic review, the relevant literature on the microbiome of peri-implantitis biofilm, comparing its composition with that of biofilms collected from healthy implant sites and periodontitis sites, was reviewed. The meta-analyses were performed to assess the prevalence of *A. actinomycetemcomitans*, *P. intermedia,* and *T. forsythia*, revealing generally higher chances of occurrence in peri-implantitis biofilms as compared to either periodontitis samples or healthy implants. Nevertheless, after taking diverse microbiological methods described in the selected studies into consideration, the presence of a characteristic peri-implantitis-related microbiome could not be confirmed, as the majority of reports did not show a substantial and consistent difference in the prevalence of specific microbial species between health and disease. Notably, several studies emphasized on the detection frequency of single microbial species, again without showing any consistency when comparing either all studies or studies using the same methods for microbial identification and qualification. Therefore, our main hypothesis, suggesting that there are specific microbial species that would exclusively, or at least predominantly, reside in the biofilm of implants with peri-implantitis only, was not confirmed due to absence of consistent evidence. Generally, there are some inconsistent indications from single studies that peri-implantitis might have a broader microbial spectrum than periodontitis, while studies describing culture-dependent techniques did not confirm these findings. Furthermore, the outcomes from hybridization techniques highlighted the presence of equally distributed taxa in health and disease. All PCR-based results failed to detect *E. nodatum* at healthy sites while *Fusobacterium* spp., *Staphylococcus* spp., *Veillonella* spp. [66], and *Treponema* spp. [98] were not detected at healthy sites using pyrosequencing. In another report on metatranscriptomic analysis, *Gemella* spp. and *Veillonella* spp. could not be identified in peri-implantitis samples [87]. Regarding viruses, however, consistently high counts of EBV and human cytomegalovirus (HCMV) were recovered from peri-implantitis biofilms [65,91,99].

A recent retrospective study analyzed the microbiological composition of about 500 biofilm samples from implants and their adjacent periodontium ten years after implant loading, using PCR amplification and hybridization [100]. As a result, a broad range of congruence (6.2%–78.4%) was measured in samples from peri-implant and periodontal pockets. In peri-implant samples, higher levels of *T. forsythia*, *P. micra*, *F. nucleatum*, *Fusobacterium necrophorum*, and *C. rectus* were found as compared to periodontitis sites. After comparing inflamed peri-implant sites with healthy ones, a higher prevalence of *P. intermedia*, *T. denticola*, *C. rectus,* and *Staphylococcus warneri* was found at inflamed sites. Smokers were characterized by a higher microbial diversity than non-smokers both in health and disease. The results of this retrospective study are in accordance with the findings of the present meta-analysis regarding both the absence of any specificity of the peri-implantitis microbiome and the high prevalence of *P. intermedia*. The submucosal prevalence of *P. intermedia* in diseased peri-implant sites versus healthy sites was confirmed in a further case-control study [58].

In general, a considerable heterogeneity of the selected studies regarding the microbial analysis method and study design, especially in terms of retrieval from comparative samples from either different cohorts or from the same individual, was found. Results obtained from different microbiological methods do not allow for a direct comparison of the respective outcomes, even though each technique has its specific advantages and disadvantages. While culture-dependent techniques were applied until the early 2000s, modern techniques based on specifying DNA sequences have allowed for more comprehensive microbiological screening in the last two decades. Additionally, other sophisticated techniques, such as pyrosequencing, allowed for the assessment of a much broader range of oral microbiota both in health and disease. Since different microbiological techniques aim also at different targets, i.e., the multiplication of vital bacteria, the detection of few “key pathogens” or the broadest possible range of analysis, a direct comparison between results from different microbiological techniques, is not indicated. Accordingly, the microbiome of peri-implantitis is characterized by a high microbial diversity consisting of both Gram-positive aerobe rods, Gram-negative anaerobic, and fusiform pathogens.

Moreover, reports on culture-based techniques did not reveal any specificity for the composition of biofilms from samples of peri-implantitis as compared to periodontitis or healthy implants, with the exception of the recovery of *C. albicans* only from healthy implants. Several studies support the existence of a higher microbial diversity in samples from healthy sites. Microbiological techniques that use specific DNA sequences, such as hybridization or PCR analysis, likewise, did not reveal a specificity of any species in health or disease. Only *P. nigrescens* and *E. nodatum* were identified more frequently in samples from peri-implantitis. Likewise, single studies indicated a high association of sub-clusters of *Treponema* spp. and *Synergistetes* spp. with peri-implantitis [30].

The decisive advantage of pyrosequencing is the ability to assess the entire microbiome as registered in the relevant libraries, including non-cultivable microbial species and, thereby, to comprehensively characterize the biofilm composition [66,101]. According to the results of studies using pyrosequencing, all tested species were positively tested in samples from healthy implants, peri-implantitis and periodontitis biofilms, with the only exception of *Campylobacter* spp. and *Gemella* spp., which were absent in samples from healthy implants in only one study [69].

No specific genera were found for the different groups. Generally speaking, if a species could not be found in either health or disease, in most cases it was also not detectable in the other respective state of health. Nevertheless, there are single studies that report a higher microbial diversity in general. More specifically, the prevalence of *Bacteroidetes* spp. [102], *Actinomyces* spp., *Peptococcus* spp., *Campylobacter* spp., *Streptococcus* spp., and *Butyrivibrio* spp. [56] in peri-implantitis than in healthy implant sites has been highlighted. Another report demonstrated higher counts of *P. nigrescens* and lower counts of *Peptostreptococcaceae* spp. and *Desulfomicrobium orale* in peri-implantitis samples than periodontitis sites. However, it is noteworthy that other reports assessing the same microbial species reported controversial results [55,90] for the prevalence of several bacterial species, namely *Prevotella* spp., *Staphylococcus* spp., *Tannerella* spp., and *Treponema* spp. Interestingly, the only study performing meta-transcriptomic assessment failed to detect *Gemella* spp. and *Veillonella* spp. in peri-implantitis samples, implying the absence of specificity of the according species.

Another study explicitly investigated the presence of *Archaea* spp. by PCR [31]. The detection frequency of *Archaea* spp. turned out to be slightly more frequent in peri-implantitis than in healthy sites. Due to the moderate difference between health and disease, however, archaea may not serve as a key species in the etiology of peri-implantitis. The presumable behavior of archaea as environmental modifiers in periodontal and peri-implant disease needs further investigation. Consistently, EBV was more frequent in peri-implantitis samples in three studies using PCR, indicating a potential role of the virus in the pathogenesis of peri-implantitis. [65,91,99]

All selected reports described standardized conventional microbiologic methods and reported exact (81%) or at least satisfying criteria (19%) for the diagnosis of peri-implantitis. This was confirmed by marginal bone loss on radiographs and pocket depths of more than 4–5 mm, absence of bleeding-on-probing or validated progression as an important indicator for the actual peri-implant inflammation. Quality assessment revealed a relatively high number of studies of low quality (48%). Notably, this outcome does not result from an inaccurate assessment or data presentation, but from the fact that solely peri-implantitis samples without controls from healthy or periodontally affected sites were considered. An adequate matching of the cohorts of different groups and control of confounders was performed in 43% of studies. Only 7% of the retrieved studies reported blinding during microbiological tests with regard to the group allocation.

Keeping these points in mind, it is obvious that the onset and progression of peri-implantitis is certainly not merely dependent on the composition of the related biofilms, but might depend on risk factors (e.g., smoking), and diseases affecting general health (e.g., diabetes mellitus), which have been revealed by a broad body of evidence [103]. Furthermore, other factors that are more difficult to assess are also likely to influence peri-implant health, e.g., psychic stress and nutrition. The latter are known to substantially influence the course of other chronic and especially inflammatory diseases [104,105].

The evaluation of the publication date of the retrieved studies revealed that first reports were published in the 1990s, while there was an abrupt rise in numbers of publications starting from the year 2010. Culture-dependent techniques have been predominantly used until the beginning of 2000, since PCR and hybridizations techniques were established at that time. More sophisticated microbiological techniques, such as pyrosequencing, have been developed since 2012, while metatranscriptomics have been applied since 2016. Each of these techniques constitutes a valid assessment method with specific emphasis on either complex determination of the whole microbiome, quick and less expensive assessment, or detection of probably unknown microbial species. A comparison of the results from studies on different microbiological techniques, however, is problematic due to the different questions that might be answered by the different techniques. Yet, a comparative report on oral microbiota revealed that results from cultivation and hybridization do not seem to perfectly match with each other [53]. The finding that the microbial profile of peri-implantitis does not display consistent characteristics is in accordance with a literature review by Lafaurie et al. [29].

## Figures and Tables

**Figure 1 microorganisms-08-00661-f001:**
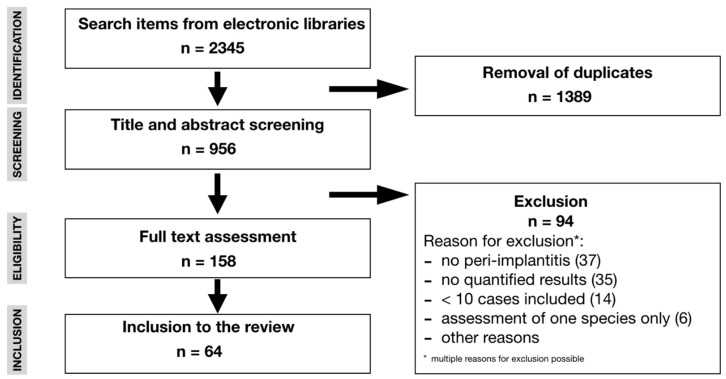
Search strategy.

**Figure 2 microorganisms-08-00661-f002:**
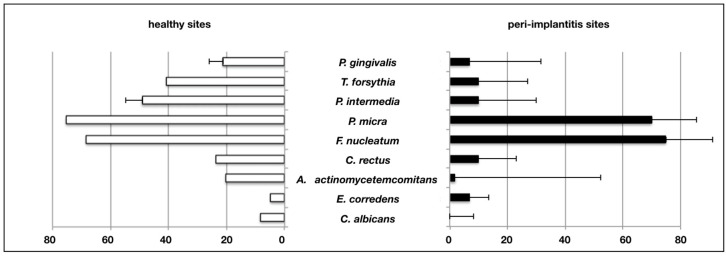
Frequency of positive detection in studies performing culture-based assessment. Weighted percentages of positive assessment in samples from healthy and peri-implantitis sites (bars = means, whiskers = standard deviation).

**Figure 3 microorganisms-08-00661-f003:**
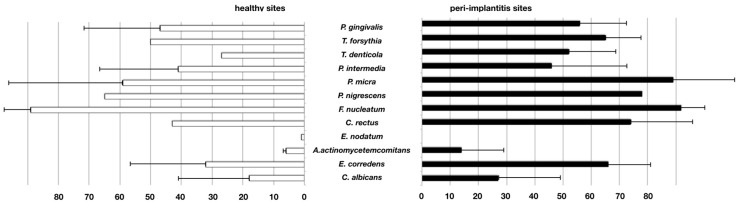
Frequency of positive detection in studies performing hybridization-based assessment. Weighted percentages of positive assessment in samples from healthy and peri-implantitis sites (bars = means, whiskers = standard deviation).

**Figure 4 microorganisms-08-00661-f004:**
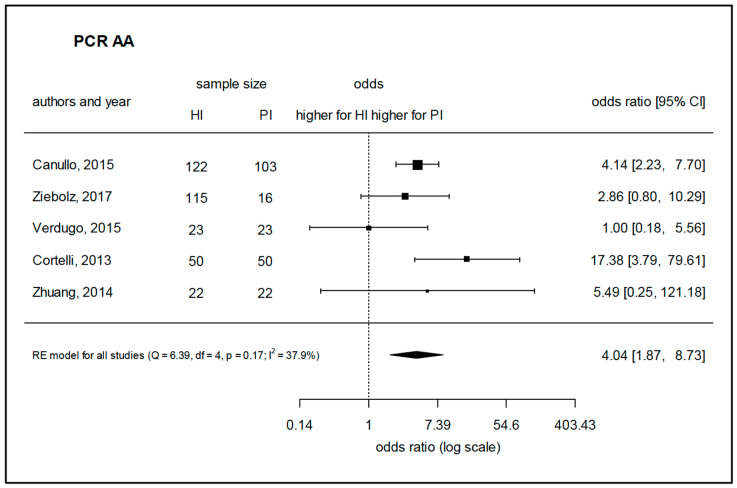
Meta-analysis for the presence of *Aggregatibacter actinomycetemcomitans* in either healthy sites or peri-implantitis sites (studies performing PCR-based analysis). HI—healthy implants, PI—implants with peri-implantitis, Q—Cochrane’s Q, df—degrees of freedom, *p*—level of significance, I^2^—proportion of observed variance (heterogeneity).

**Figure 5 microorganisms-08-00661-f005:**
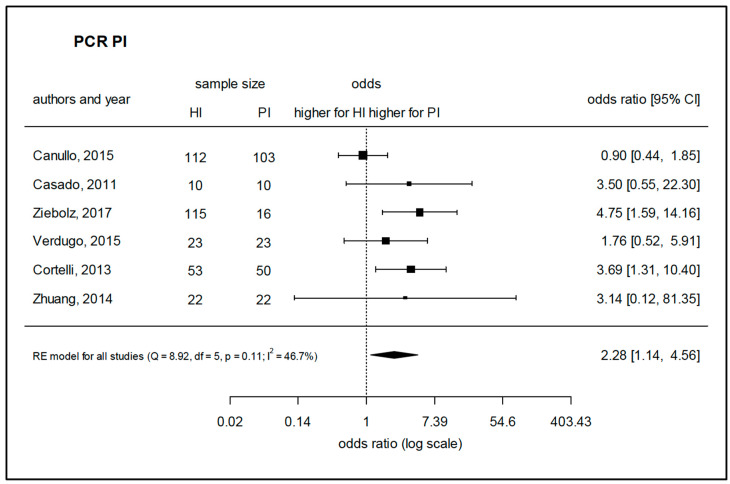
Meta-analysis for the presence of *Porphyromonas gingivalis* in either healthy sites or peri-implantitis sites (studies performing PCR-based analysis). HI—healthy implants, PI—implants with peri-implantitis, Q—Cochrane’s Q, df—degrees of freedom, *p*—level of significance, I^2^—proportion of observed variance (heterogeneity).

**Figure 6 microorganisms-08-00661-f006:**
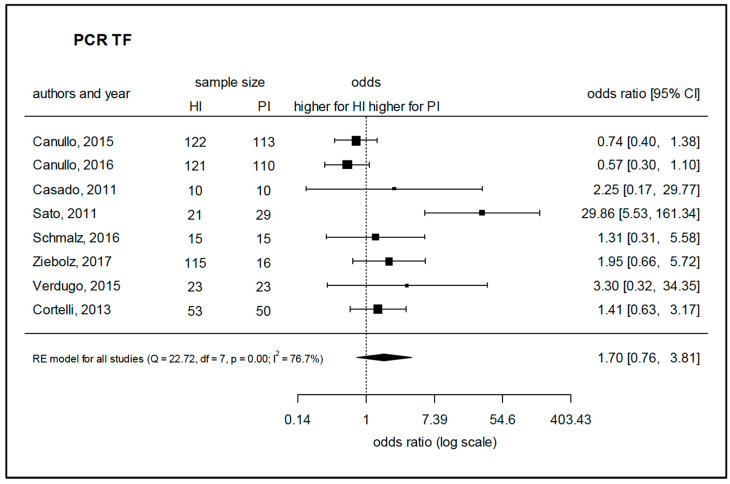
Meta-analysis for the presence of *Tannerella forsythia* in either healthy sites or peri-implantitis sites (studies performing PCR-based analysis). HI—healthy implants, PI—implants with peri-implantitis, Q—Cochrane’s Q, df—degrees of freedom, *p*—level of significance, I^2^—proportion of observed variance (heterogeneity).

**Figure 7 microorganisms-08-00661-f007:**
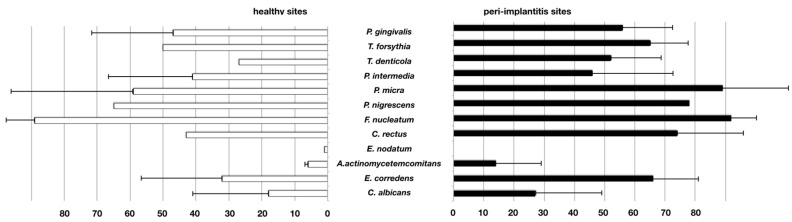
Frequency of positive detection in studies performing PCR-based assessment. Weighted percentages of positive assessment in samples from healthy and peri-implantitis sites (bars = means, whiskers = standard deviation).

**Table 1 microorganisms-08-00661-t001:** Quality assessment.

Author and Date of Publication	Case Definition	Selection of Controls	Matched for Bias	Assessment of Outcome	Score	Quality
Albertini et al., 2015 [36]	2	2	0	0	4	Medium
Alcoforado 1990 [37]	1	0	0	0	1	Low
Arisan et al., 2015 [38]	2	0	0	0	2	Low
Ata-Ali et al., 2015 [39]	2	1	1	0	4	Medium
Augthun et al., 1997 [40]	1	0	0	0	1	Low
Bassetti et al., 2014 [41]	2	0	0	0	2	Low
Becker et al., 1990 [42]	1	0	0	0	1	Low
Bertone et al., 2016 [43]	2	0	0	0	2	Low
Birang et al., 2017 [44]	2	0	0	0	2	Low
Bombeccari 2013 [45]	2	0	0	0	2	Low
Botero et al., 2005 [46]	2	1	2	0	5	Medium
Caccianiga et al., 2016 [47]	1	0	0	0	1	Low
Canullo et al., 2013 [48]	2	1	2	0	5	Medium
Canullo et al., 2015a [49]	2	0	0	0	2	Low
Canullo et al., 2015b [50]	2	1	2	0	5	Medium
Canullo et al., 2017 [51]	2	1	2	0	5	Medium
Casado et al., 2011 [52]	1	1	2	0	4	Medium
Charalampakis et al., 2012 [53]	2	0	0	0	2	Low
Cortelli et al., 2013 [54]	2	2	2	0	6	High
da Silva et al., 2014 [55]	2	1	2	0	5	Medium
Dabdoub et al., 2013 [56]	2	2	2	0	6	High
de Waal et al., 2014 [57]	2	0	0	0	2	Low
de Waal et al., 2017 [58]	2	1	2	0	5	Medium
Dörtbudak et al., 2011 [59]	2	0	0	0	2	Low
Ebadian et al., 2012 [60]	2	1	2	1	6	High
Faveri et al., 2011 [61]	2	1	2	0	5	medium
Gurlek et al., 2017 [62]	2	2	2	0	6	high
Hultin et al., 2002 [63]	1	1	2	0	4	medium
Isehed et al., 2016 [64]	2	0	0	0	2	low
Jankovic et al., 2011 [65]	2	1	2	0	5	medium
Kumar et al., 2012 [66]	2	1	2	0	5	medium
Leonhardt et al., 1999 [67]	2	1	2	0	5	medium
Leonhardt et al., 2003 [68]	2	0	0	0	2	low
Maruyama et al., 2014 [69]	2	2	2	0	6	high
Maximo et al., 2009 [70]	2	1	2	1	6	high
Mombelli et al., 2001 [71]	2	0	0	0	2	low
Neilands et al., 2015 [72]	2	1	1	1	5	high
Parthiban et al., 2017 [73]	2	1	1	0	4	medium
Passariello et al., 2012 [74]	1	0	0	0	1	low
Persson et al., 2006 [75]	1	0	0	0	1	low
Persson et al., 2010 [76]	2	0	0	0	2	low
Persson et al., 2011 [77]	2	0	0	0	2	low
Renvert et al., 2006 [78]	2	0	0	0	2	low
Renvert et al., 2007 [79]	2	1	2	0	5	medium
Renvert et al., 2009 [80]	2	0	0	0	2	low
Renvert et al., 2015 [10]	1	0	0	0	1	low
Renvert et al., 2017 [81]	2	0	0	0	2	low
Salcetti et al., 1997 [82]	1	1	1	1	4	medium
Sato et al., 2011 [83]	2	0	0	0	2	low
Sbordone et al., 1995 [84]	2	0	0	0	2	low
Schmalz et al., 2016 [85]	2	1	1	0	4	medium
Schwarz et al., 2015 [86]	2	1	1	0	4	medium
Shiba et al., 2016 [87]	2	2	2	0	6	high
Shibli et al., 2008 [88]	2	1	2	0	5	medium
Tada et al., 2017 [89]	2	0	0	0	2	low
Tsigarida et al., 2015 [90]	1	1	1	0	3	medium
Verdugo et al., 2015 [91]	2	2	2	0	6	high
Zhuang et al., 2016 [92]	2	2	2	0	6	high
Ziebolz et al., 2017 [93]	2	1	2	0	5	medium

**Table 2 microorganisms-08-00661-t002:** Studies assessing different species using culture-based techniques.

Author and Date of Publication	Assessed Situation	*P. gingivalis*	*T. forsythia*	*P. intermedia*	*P. micra*	*F. nucleatum*	*C. rectus*	*A. actinomycetemcomitans*	*E. corrodens*	*C. albicans*	*n*
**Studies assessing only peri-implantitis-affected sites**											
Alcoforado et al., 1990 [37]	PI	x	x	22%	33%	x	x	6%	x	28%	18
Augthun et al., 1997 [40]	PI	x	x	28%	x	22%	x	89%	17%	x	18
Bombeccari et al., 2013 [45]	PI	100%	x	100%	x	x	x	78%	x	x	40
Charalampakis et al., 2012 [53]	PI	9%	x	52%	x	x	x	13%	x	3%	281
de Waal et al., 2013 [57]	PI	15%	21%	14%	69%	50%	15%	0%	x	x	78
Dortbudak et al., 2011 [59]	PI	100%	x	100%	x	x	x	100%	x	x	15
Leonhardt et al., 2003 [68]	PI	4%	x	38%	x	x	x	54%	x	0%	26
Mombelli et al., 2001 [71]	PI	17%	37%	55%	x	x	56%	0%	0%	x	30
Passariello et al., 2012 [74]	PI	x	x	x	x	x	x	x	x	x	128
**Studies comparing healthy sites with peri-implantitis-affected sites**											
Botero et al., 2005 [46]	PI	44%	x	25% *	x	x	x	x	0%	x	16
	HI	0%	x	7% *	x	x	x	x	7%	x	15
de Waal et al., 2017 [58]	PI	25%	60%	50%	90%	95%	20%	4%	x	x	85
	HI	12%	10%	5%	70%	75%	10%	2%	x	x	69
Leonhardt et al., 1999 [67]	PI	8%	x	60%	x	x	x	27%	x	8%	37
	HI	2%	x	18%	x	x	x	2%	x	0%	51
Neilands et al., 2015 [72]	PI	x	x	x	x	x	x	x	x	x	25
	HI	x	x	x	x	x	x	x	x	x	25

HI—healthy implants, PI—implants with peri-implantitis, x not assessed, *n*—number of samples assessed for the respective study. * No differentiation of *P. intermedia* and *P. nigrescens*.

**Table 3 microorganisms-08-00661-t003:** Studies assessing different species using PCR-based techniques.

Author and Year of Publication	Assessed Situation	*P. gingivalis*	*T. forsythia*	*T. denticola*	*P. intermedia*	*P. micra*	*P. nigrescens*	*F. nucleatum*	*C. rectus*	*E. nodatum*	*A. actinomycetemcomitans*	*E. corrodens*	*C. albicans*	*n*
**Study comparing peri-implantitis sites with periodontitis sites**														
Albertini et al., 2015 [36]	PI	66	75	39	24	x	x	x	x	x	0	x	x	33
	PER	48	60	27	27	x	x	x	x	x	0	x	x	33
**Studies comparing healthy sites with peri-implantitis-affected sites**														
Ata-Ali et al., 2015 [39]	PI	37	33	33	x	x	x	x	x	x	x	x	x	24
	HI	11	17	17	x	x	x	x	x	x	x	x	x	54
Canullo et al., 2015 [49]	PI	57	75	65	75	97	x	99	86	x	0	75	17	113
	HI	67	80	43	77	97	x	97	77	x	3	53	10	122
Casado et al., 2011 [52]	PI	80	90	70	70	x	x	x	x	x	40	x	x	10
	HI	90	80	70	40	x	x	x	x	x	30	x	x	10
Sato et al., 2011 [83]	PI	86	76	76	x	x	x	x	x	x	14	x	x	29
	HI	5	10	10	x	x	x	x	x	x	0	x	x	21
Schmalz et al., 2016 [85]	PI	67	60	47	x	x	x	93	x	x	20	x	x	15
	HI	80	53	27	x	x	x	93	x	x	13	x	x	15
Ziebolz et al., 2017 [93]	PI	69	62	50	50	31	x	93	6	0	25	6	x	16
	HI	63	46	27	17	16	x	83	4	1	10	8	x	115
Bertone et al., 2016 [43]	PI	x	x	x	x	x	x	x	x	x	x	x	83	18
	HI	x	x	x	x	x	x	x	x	x	x	x	59	22
Schwarz et al., 2015 [86]	PI	x	x	x	x	x	x	x	x	x	x	x	32	19
	HI	x	x	x	x	x	x	x	x	x	x	x	40	10
Verdugo et al., 2015 [91]	PI	61	96	78	44	x	78	x	x	x	13	x	x	23
	HI	65	87	35	30	x	65	x	x	x	13	x	x	23
Kato et al., 2017 [94]	PI	60	x	x	x	x	x	x	x	x	x	x	x	15
	HI	93	x	x	x	x	x	x	x	x	x	x	x	15
**Studies assessing peri-implantitis sites only**														
Arisan et al., 2015 [38]	PI	77	92	100	83	92	88	100	83	0	x	x	x	48
Bassetti et al., 2014 [41]	PI	38	60	45	30	68	x	95	58	55	35	55	x	40
Becker et al., 1990 [42]	PI	38	x	x	35	x	x	x	x	x	28	x	x	36
Caccianiga et al., 2016 [47]	PI	90	80	70	x	x	x	80	100	x	10	80	x	10
**Studies comparing peri-implantitis sites with periodontitis sites and healthy implants**														
Cortelli et al., 2013 [54]	PI	54	40	54	32	x	x	x	70	x	42	x	x	50
	PER	72	100	76	56	x	x	x	98	x	4	x	x	50
	HI	6	32	12	12	x	x	x	98	x	4	x	x	53
Zhuang et al., 2016 [92]	PI	32	x	18	5	x	x	77	x	x	9	x	x	22
	PER	32	x	23	5	x	x	77	x	x	14	x	x	22
	HI	23	x	14	0	x	x	64	x	x	0	x	x	22

PI—implants with peri-implantitis, HI—healthy implants, PER—teeth with periodontitis, x not assessed, *n*—number of samples assessed for the respective study.

**Table 4 microorganisms-08-00661-t004:** Studies assessing different species using hybridization techniques.

Author and Year of Publication	Assessed Situation	*Actinomyces* spp.	*Campylobacter* spp.	*Fusobacterium* spp.	*Gemella* spp.	*Parvimonas* spp.	*Porphyromonas* spp.	*Prevotella* spp.	*Staphylococcus* spp.	*Streptococcus* spp.	*Tannerella* spp.	*Treponema* spp.	*Veillonella* spp.	*n*
**Studies comparing healthy sites with peri-implantitis-affected sites**														
Hultin et al., 2002 [63]	PI	x	71	100	x	x	65	100	x	88	x	41	x	17
	HI	x	64	100	x	x	79	93	x	71	x	43	x	14
Maximo et al., 2009 [70]	PI	+	+	+	+	+	+	+	x	+	+	+	+	13
	HI	+	-	+	+	+	-	+	-	+	+	+	+	10
Renvert et al., 2007 [79]	PI	5.9	8.7	44.5	x	x	x	18	13	4.7	4.3	8.7	13	213
	HI	2.9	3.7	44. 7	x	x	x	26.3	7.8	2.6	2.6	13.2	18.4	213
Shibli et al., 2008 [88]	PI	+	+	+	+	+	+	+	x	+	+	+	+	22
	HI	x	+	+	+	+	x	+	x	+	+	x	+	22
Ata-Ali et al., 2015 [39]	PI	x	x	x	x	x	37.5	x	x	x	33.3	33.3	x	24
	HI	x	x	x	x	x	11.1	x	x	x	22.2	16.7	x	54
Salcetti et al., 1997 [82]	PI	+	+	69.6	x	+	+	87	x	47.8	+	+	x	21
	HI	+	+	14.3	x	+	+	28.6	x	0				8
**Studies comparing peri-implantitis sites with periodontitis sites and healthy implants**														
Ebadian et al., 2012 [60]	PI	+	15.4	38.5	x	x	53.8	30.8	x	x	61.5	x	x	13
	PER	+	0	61.5	x	x	30.8	30.8	x	x	46.1	x	x	13
	HI	+	59.1	72.7	x	x	90.9	90.9	x	x	90.9	x	x	21
**Studies assessing only peri-implantitis**														
Charalampakis et al., 2012 [53]	PI	+	+	+	x	+	+	+	x	+	+	+	x	257
Persson et al., 2006 [75]	PI	+	x	+	+	+	+	+	x	−	+	+	x	25
Persson et al., 2011 [77]	PI	+	+	+	x	+	+	+	+	+	+	+	+	37
Renvert et al., 2006 [78]	PI	+	+	+	x	x	+	+	+	x	+	+	x	32
Renvert et al., 2008 [26]	PI	+	+	+	x	+	+	+	x	+	+	+	x	37
Renvert et al., 2015 [95]	PI	+	+	+	x	x	x	+	+	+	+	x	x	41
Renvert et al., 2017 [81]	PI	+	+	+	x	+	x	x	+	+	+	x	x	41

PI—implants with peri-implantitis, HI—healthy implants, PER—teeth with periodontitis, spp.—species, + positive detection, - no detection, x not assessed, *n*—number of samples assessed for the respective study.

**Table 5 microorganisms-08-00661-t005:** Studies assessing different species using pyrosequencing.

Author and Year of Publication	Assessed Situation	*Actinomyces* spp.	*Campylobacter* spp.	*Fusobacterium* spp.	*Gemella* spp.	*Parvimonas* spp.	*Porphyromonas* spp.	*Prevotella* spp.	*Rothia* spp.	*Staphylococcus* spp.	*Streptococcus* spp.	*Tannerella* spp.	*Treponema* spp.	*Veillonella* spp.	*n*
**Studies comparing healthy sites with peri-implantitis-affected sites**														
Da Silva et al., 2014 [55]	PI	+	+	+	+	+	+	-	+	-	-	-	-	+	10
	HI	+	+	+	+	+	+	-	+	-	-	-	-	+	10
Tsigarida et al., 2015 [90]	PI	+	+	+	+	+	+	+	+	+	+	+	+	+	40
	HI	+	+	+	+	+	+	+	+	+	+	+	+	+	20
Zheng et al., 2015 [97]	PI	+	x	+	+	x	x	x	x	x	x	x	+	+	6
	HI	+	x	-	+	x	x	x	x	x	x	x	-	+	10
**Studies comparing healthy sites with peri-implantitis-affected and periodontitis-affected sites**														
Dabdoub et al., 2013 [56]	PI	+	+	+	+	-	+	+	+	+	+	+	+	+	20
	HI	+	+	+	+	-	+	+	+	+	+	+	+	+	12
	PAR	+	+	+	+	-	+	+	+	+	+	+	+	+	17
Kumar et al., 2012 [66]	PI	+	+	+	+	+	+	+	+	+	+	-	+	+	10
	HI	+	+	-	+	+	+	+	+	-	+	-	+	-	10
	PER	+	+	+	+	+	+	+	+	+	+	-	+	+	10
**Study comparing peri-implantitis sites with periodontitis sites**														
Maruyama et al., 2014 [69]	PI	+	-	+	-	+	+	+	+	+	+	+	+	+	20
	PER	+	-	+	-	+	+	+	+	+	+	+	+	+	20
Shiba et al., 2016 [87]	PI	+	+	-	-	+	+	+	+	-	+	+	+	-	12
	PER	+	+	-	+	+	+	+	+	-	+	+	+	+	12

PI—implants with peri-implantitis, HI—healthy implants, PER—teeth with periodontitis, spp. —species, + positive detection, - no detection, x not assessed, *n*—number of samples assessed for the respective study.

**Table 6 microorganisms-08-00661-t006:** Studies assessing different species using metatranscriptomic analysis.

Author and Year of Oublication	Assessed Situation	*Actinomyces* spp.	*Campylobacter* spp.	*Fusobacterium* spp.	*Gemella* spp.	*Parvimonas* spp.	*Porphyromonas* spp.	*Prevotella* spp.	*Rothia* spp.	*Staphylococcus* spp.	*Streptococcus* spp.	*Tannerella* spp.	*Treponema* spp.	*Veillonella* spp.	*n*
**Study comparing peri-implantitis sites with periodontitis sites**														
Shiba et al., 2016 [87]	PI	+	+	-	-	+	+	+	+	-	+	+	+	-	12
	PER	+	+	-	+	+	+	+	+	-	+	+	+	+	12

PI—implants with peri-implantitis, PER—teeth with periodontitis, spp. —species, + positive detection, - no detection, *n*—number of samples assessed for the respective study.

**Table 7 microorganisms-08-00661-t007:** Studies specifically assessing *S. aureus.*

Author and Year of Publication	Positive Assessment of *S. aureus* [%]		*n*
**Cultivation-based assessment**			
Charalampakis et al., 2012 [53]	PI	8.9	281
Leonhardt et al., 1999 [67]	PI	0	37
	HI	0	51
Leonhardt et al., 2003 [68]	PI	11	9
Passariello et al., 2012 [74]	PI	18.75%	62.5

**PCR-based assessment**			
Zhuang et al., 2016 [92]	PI	77.3	22
	HI	72.7	22
	PER	68.2	22

PI—implants with peri-implantitis, HI—healthy implants, PER—teeth with periodontitis, spp.—species, *n*—number of samples assessed for the respective study.

**Table 8 microorganisms-08-00661-t008:** Studies assessing archaea and virus.

Author and Date of Publication		Archaea	Virus			*n*
			HCMV (HCMV1/HCMV2)	EBV (EBV1/EBV2)	HSV	
**Cultivation-based assessment**						
Faveri et al., 2011 [61]	PI	48	x	x	x	25
	HI	12	x	x	x	25
Jankovic et al., 2011a [65]	PI	x	65	45	x	20
	HI	x	6	11	x	18
Jankovic et al., 2011b [99]	PI	x	33/53	7/10	x	30
	HI	x	8/0	4/8	x	25
Parthiban et al., 2017 [73]	PI	x	x	x	33	77
	HI	x	x	x	24	113
Verdugo et al., 2015 [91]	PI	x	4	39	x	23
	HI	x	4	17	x	23

PI—implants with peri-implantitis, HI—healthy implants, x not assessed, *n*—number of samples assessed for the respective study, HCMV—human cytomegalovirus, EBV—Epstein–Barr virus, HSV—herpes simplex virus.

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
