# Peer review of "The Microbiome of Peri-Implantitis: A Systematic Review and Meta-Analysis"

_microorganisms, 2020, doi:10.3390/microorganisms8050661_

Round 1
Reviewer 1 Report
The systematic review entitled " The Microbiome of Peri-Implantitis. A Systematic Review and Meta-Analysis” deals with the comparison of microbial profiles of peri-implantitis to those of periodontitis and healthy implants.
The study is very interesting; however, some minor changes need to be addressed.
Keywords: should be in alphabetical order, KEYWORDS should not contain the same words that are within the title of the text. Thus these should be changed appropriately.
Introduction
Describe in this section possible methods/devices of treating peri-implantitis e.g., scalers, lasers, air-abrasive instruments. Add the recent citation to this statement: „Accordingly, any cause-related treatment of peri-implantitis aims at effectively removing established microbial biofilms and preventing new biofilm formation“ https://www.mdpi.com/2076-2607/7/7/189
Add the recent citation to this statement: „In particular, mechanical access for debridement of submucosal areas and narrow infra-bony defects is difficult and cannot be performed to a satisfactory extent“ https://doi.org/10.3390/ma12223748
Author Response
Referee(s)' Comments to Author:
Reviewer 1
Comments to the Author
The systematic review entitled " The Microbiome of Peri-Implantitis. A Systematic Review and Meta-Analysis” deals with the comparison of microbial profiles of peri-implantitis to those of periodontitis and healthy implants.
The study is very interesting; however, some minor changes need to be addressed.
Response: The authors would like to thank the reviewer for the kind appreciation and the constructive criticism.
- Keywords: should be in alphabetical order, KEYWORDS should not contain the same words that are within the title of the text. Thus these should be changed appropriately.
Response: The referee is right. Accordingly, we eliminated “microbiome” and peri-implantitis” from the keywords, added “oral pathogens” and changed the sequence of the keywords according to the alphabetic order.
Revised text section:
culture-dependent techniques, hybridization, oral pathogens, PCR, pyrosequencing
- Introduction
Describe in this section possible methods/devices of treating peri-implantitis e.g., scalers, lasers, air-abrasive instruments.
Response: The authors thank the referee for this valuable suggestion. We added a paragraph on the treatment options for peri-implantitis including the instruments, which are typically used.
Revised text section:
Therefore, hand instruments like scalers or curettes, ultrasonic tips or air-abrasive devices [6,17] and sometimes different kind of lasers [18] are used.
- Add the recent citation to this statement: „Accordingly, any cause-related treatment of peri-implantitis aims at effectively removing established microbial biofilms and preventing new biofilm formation“ https://www.mdpi.com/2076-2607/7/7/189
Response:
Again, the authors than for the valuable recommendation. We found the study worth a citation and decided to better place it together with the other instruments usually employed for peri-implantitis therapy.
Revised text section:
(please see answer to comment 2, citation nr 18)
- Add the recent citation to this statement: „In particular, mechanical access for debridement of submucosal areas and narrow infra-bony defects is difficult and cannot be performed to a satisfactory extent“ https://doi.org/10.3390/ma12223748
Response: Again, the citation was added as suggested.
Revised text section:
In particular, mechanical access for debridement of submucosal areas and narrow infra-bony defects is difficult and cannot be performed to a satisfactory extent [17,21,22].
Reviewer 2 Report
The present manuscript describe a systematic review and meta-analysis of the microbiome of peri-implantitis.
The review is very well described and detailed, although the conclusion can not really differ the microbiome of peri-implantitis of the microbiome of periodontitis. The reason I see for that is that the environment impose a great factor on the microbial profiles of microbiomes in general. I understand that, for treatment reasons, the knowledge on the microbial population in different diseases would be of great interest, but we should not forget that other factors may interfere in a biofilm formation and selection of microbes.
When we take the gut as example we know that nutrition can interfere in the bacterial population of its microbiome. Its is known that according to what an individual eat in a day, can alter the bacterial population (not only quantitatively but also qualitatively) in question of hours (doi: 10.1186/s12967-017-1175-y). Could be environment factors, as for example, nutrition, also important for the quality of a peri-implantitis and periodontitis microbiome as well? The authors describe some findings about the smoke habit in patients and its influences on the oral microbiome.
In my opinion, even if the authors here don't enrich the review with data about it, at least it should be discussed as a possible explanation for the heterogeneity of the results obtained.
Author Response
Reviewer 2
Comments to the Author
- The present manuscript describe a systematic review and meta-analysis of the microbiome of peri-implantitis.
The review is very well described and detailed, although the conclusion can not really differ the microbiome of peri-implantitis of the microbiome of periodontitis. The reason I see for that is that the environment impose a great factor on the microbial profiles of microbiomes in general. I understand that, for treatment reasons, the knowledge on the microbial population in different diseases would be of great interest, but we should not forget that other factors may interfere in a biofilm formation and selection of microbes.
When we take the gut as example we know that nutrition can interfere in the bacterial population of its microbiome. Its is known that according to what an individual eat in a day, can alter the bacterial population (not only quantitatively but also qualitatively) in question of hours (doi: 10.1186/s12967-017-1175-y). Could be environment factors, as for example, nutrition, also important for the quality of a peri-implantitis and periodontitis microbiome as well? The authors describe some findings about the smoke habit in patients and its influences on the oral microbiome.
In my opinion, even if the authors here don't enrich the review with data about it, at least it should be discussed as a possible explanation for the heterogeneity of the results obtained.
Response: The authors agree with the referee to the full extent. Indeed, the study results and its conclusion reveal that the chase for a special composition of the peri-implant microbiome is misleading and not reasonable. The authors thank the referee for his/her input and we included the illustrative execution in part within the discussion section of the revised manuscript.
Revised text section:
Keeping these points in mind, it is obvious that the onset and progression of peri-implantitis is certainly not merely dependent on the composition of the related biofilms, but might depend on risk factors like smoking, diseases affecting general health like diabetes mellitus, which have been revealed by a broad body of evidence [101]. Furthermore, other factors that are more difficult to assess are also likely to influence peri-implant health, e.g. psychic stress and nutrition. The latter are known to substantially influence the course of other chronic and especially inflammatory diseases [102,103].